# Efficacy of Three Commercially Available Desensitizers in Reducing Post-Operative Sensitivity Following Composite Restorations: A Randomized Controlled Clinical Trial

**DOI:** 10.3390/polym14071417

**Published:** 2022-03-30

**Authors:** Mohammed E. Sayed, Harisha Dewan, Rawabi Kharaf, Maram Athlawi, Munira Alfaifi, Maryam Hassan Mugri, Razan Abu-Alqasem Bosly, Nada Yousef Fageehi, Maryam Hadi, Bayan Jebril Zurbtan, Fawzia Ibraheem Shaabi, Fatimah H. Alsurayyie, Dalea Mohammed Bukhary, Ruwaida Zaki Alshali, Hitesh Chohan

**Affiliations:** 1Department of Prosthetic Dental Sciences, College of Dentistry, Jazan University, Jazan 45142, Saudi Arabia; fshaabi@jazanu.edu.sa (F.I.S.); falsoraiee@jazanu.edu.sa (F.H.A.); 2College of Dentistry, Jazan University, Jazan 45142, Saudi Arabia; rawabijaber220@gmail.com (R.K.); maram0hadi@gmail.com (M.A.); munira.alfaifi@gmail.com (M.A.); 3Department of Maxillofacial Surgery and Diagnostic Sciences, College of Dentistry, Jazan University, Jazan 45142, Saudi Arabia; dr.mugri@gmail.com; 4Jazan Specialty Dental Center, Ministry of Health, Jazan 45142, Saudi Arabia; razanbosly@gmail.com (R.A.-A.B.); nadafageehi@gmail.com (N.Y.F.); 5Primary Care Administration, Ministry of Health, Jazan 45142, Saudi Arabia; Maryam_hadi@outlook.sa; 6Hospital Administration Sector, Ministry of Health, Bisha 67714, Saudi Arabia; dr.bayanjebril@gmail.com; 7Department of Oral and Maxillofacial Prosthodontics, Faculty of Dentistry, King Abdulaziz University, Jeddah 21589, Saudi Arabia; dbukhary@kau.edu.sa (D.M.B.); ralshali@kau.edu.sa (R.Z.A.); 8Department of Restorative Dental Sciences, College of Dentistry, Jazan University, Jazan 45142, Saudi Arabia; drhiteshchohan@yahoo.co.in

**Keywords:** Gluma dentin desensitizer, Shield Force, Telio CS desensitizer, sensitivity

## Abstract

One of the most widely used esthetic restorations in dentistry is composite. The widespread application of composites can be related to advancements in biomaterials. However, due to various factors, composites are commonly associated with dental sensitivity. Hence, the present study evaluates and compares the effectiveness of three desensitizing agents in reducing post-treatment sensitivity for Class I composite restoration. Eighty subjects with Class I cavities were selected according to the inclusion criteria, and a randomized, double-blind, controlled clinical trial was carried out. Twenty patients were randomly assigned to four groups: Group C (Control group), Group GL (Gluma group), Group SF (Shield Force Plus group), and Group TC (Telio CS group). The desensitizers were applied after Class 1 cavity preparation and acid etching in all the groups, except the Control group, and thereafter, composite restoration was completed in a conventional manner. Questionnaires were provided to all the participants to record the post-operative pain/sensitivity level according to the visual analogue scale (VAS) on intake of cold drinks, intake of hot drinks, and intake of sugar for different periods of time. Significant variation was observed between the three desensitizers for all three stimuli. However, no significant variations were seen with the various age groups and between the maxillary and the mandibular teeth at the different time periods. Group GL performed better than Group SF and Group TC. It can be proposed that the application of the desensitizers reduced the post-restorative sensitivity in the composite restorations and improved acceptance.

## 1. Introduction

Esthetic materials are more readily accepted by patients for restorations. Composites have revolutionized the field of dentistry. Due to recent advances, composite restorations have been applied in the anterior, posterior, and core build-up [1,2,3]. The biocompatibility and the longevity of these restorations are also superior to those of many other restorations. However, there are a few drawbacks to composite restoration. A few patients have complained about postoperative sensitivity after the composite restorations. This might be due to microleakage injuring the pulp tissue [1]. The components present in the composite, such as monomer or acrylic groups, have shown toxicity to pulpal tissues. Moreover, there is only physical bonding between the tooth and the composite. This leaves a freeway for microorganisms and other substances from the external surfaces to slip into the interior of the tooth through cracks, due to microleakage. This has been associated with patients complaining of sensitivity to various substances that are cold, hot, or sweet. Microleakage is also associated with the instigation of secondary caries. The post-operative sensitivity associated with composite restoration is described as a sharp pain that lingers for a short period of time. It is usually provoked by stimuli such as food intake or the nature of the food, such as sweet or sour foods. This sensitivity may last for days. The success of the composite restoration is dependent on the cavity design and proper application of adhesives and additional materials that may be applied, such as liners that prevent microleakage [4,5,6,7]. These liners have also been shown to improve the bonding and decrease the post-operative sensitivity in the composites. Some liners with fluoride releasing abilities have been shown to prevent caries.

The application of dental desensitizing agents is one of the most preferred methods to prevent dental hypersensitivity [7]. Glutaraldehyde present in these desensitizers reacts with the dentinal protein component and forms a clogging mass, which decreases the tubular size or seals the dentinal tubule. The same mechanism is seen in Gluma desensitizer by Heraeus Kulzer, Hanau, Germany and Shield Force Plus by Tokuyama Dental America Inc., San Diego, CA, USA. Telio CS by Ivoclar Vivadent, Schaan, Liechtenstein is another such desensitizer that is easily available commercially. A recent study by Sayed, M.E., in 2021, stated that Gluma desensitizer was more effective for reducing sensitivity following tooth preparation when compared with Shield Force Plus and Telio CS counterpart agents [8]. Another study by Dewan, H. used the same desensitizing agents and concluded that a single application of any of these agents before cementation of zirconia crowns also increased the bond strength [9]. Hence, in the present study, we intend to evaluate and compare the effectiveness of the same three desensitizing agents in reducing the post-treatment sensitivity for Class I composite restorations. The null hypothesis tested was that the application of desensitizing agents on the prepared tooth surface for Class I cavity preparation for composite restoration has no effect in reducing post-restoration sensitivity.

## 2. Materials and Methods

The present study was designed as a single-center, parallel, double-blind, randomized, and controlled clinical trial. Ethical approval for this study was obtained from the Committee for Scientific Research, College of Dentistry, Jazan University (Ref letter no. CODJU-19722) before beginning the study. The clinical trials were registered with ClinicalTrials.gov on 26 August 2021, with the identifier no. NCT05024669.

A total of 80 patients (aged 20–45 years) who required Class I composite restorations in their molars were selected for the study, from December 2020 to April 2021. After explaining the investigation, consent was taken for the study in the form of signatures on a form. The selection criteria were
Vital teeth (electric pulp test);Apical periodontal ligament space radiographically identified;The remaining dentin thickness being at least 1 mm.

The exclusion criteria were
Restored, periodontally weak non-vital teeth.

### 2.1. Sample Size Selection

A total of 16 participants were selected for a pilot study. The results of the pilot study were used to calculate the pooled variance. A clinically significant reduction in tooth sensitivity was considered when a difference of 1 in the visual analog scale (VAS) score between the subsequent visits was detected. The following formula was used to calculate the sample size:

N = [4 × S^2^ (Z_α_ + Z_β_)^2^]/(d)^2^
where N is the sample size, S^2^ is the pooled variance, Z_α_ is the desired level of confidence, Z_β_ is the desired power, and d is the clinically expected difference. The following was the outcome:N = [4 × 0.63 (1.96 + 0.842)^2^]/(1)^2^ = 19.7=20 per group

### 2.2. Randomization

The desensitizers were assigned to the patients using simple random sampling via a lottery. The numbers were randomly selected, with each number corresponding to either a desensitizer or the control group. The three desensitizers used in the current study are shown in Figure 1.

The 4 groups, comprised of 20 patients each, were as follows (Figure 2):

Group C: no desensitizer was applied before the restoration (control group), (n = 20);

Group GL: Gluma desensitizer was applied before the restoration (n = 20);

Group SF: Shield Force Plus desensitizer was applied before the restoration (n = 20); and

Group TC: Telio CS desensitizer was applied before the restoration. (n = 20).

The manufacturer details, composition, and mechanisms of action of the desensitizers and the composite used are given in Table 1.

### 2.3. Blinding

The application of the desensitizers (or control) was performed by a single, well-trained operator. Following completion of the informed consent, each study participant was allocated a number by the operator. The patients were blinded to whether they were assigned to the control or one of the desensitizers groups. The application of the desensitizers was performed by one well-trained operator only. Therefore, the randomization process, along with the application of the desensitizing agents, was carried out by the same operator. The random allotment protocol and process were concealed from the patients who assessed and recorded the sensitivity scores, making the study double-blinded.

### 2.4. Clinical Procedures

After standard cavity preparation, the teeth were acid etched using 37% phosphoric acid. Subsequently, an air/water spray was applied for 10 s to remove the etching gel. In Group C, two layers of adhesive bonding agent were applied with a disposable brush to the entire cavity after acid etching, following the manufacturer’s recommendations. The material was light-cured for 10 s. Next, a condensable composite (Tetric N-Ceram Bulk Fill by Ivoclar Vivadent) was applied and condensed inside the cavity in a single 5-mm layer (maximum thickness) and light-cured for 40 s. The occlusion was checked. The resin surfaces were accessed, finished, and polished in the same visit.

In all 3 desensitizer groups (Group G, SF, TC), the desensitizer application was performed with a disposable brush (according to the manufacturers’ instructions) after acid etching and was maintained over the dentine for 1 min (Figure 3). The excess was removed with a gentle stream of air, and the cavity was washed and dried. The tooth was restored similarly to those in the control group.

### 2.5. Evaluation of Post-Operative Sensitivity

All patients included in the investigation, in each particular group, were given the same treatment. The patients were given a questionnaire and instructed to record any post-operative pain/sensitivity level according to the VAS for that tooth during specific periods, which were 1 day, 1 week, and 1 month after restoration, and triggered by different stimuli, such as (i) intake of cold drinks (cold stimuli), (ii) intake of hot drinks (hot stimuli), and (iii) intake of sugar (sweet stimuli).

The obtained data were subjected to statistical analysis. The dependable variable was the amount of sensitivity, and the independent variables were age, gender, and the location of teeth. The Shapiro–Wilk test was utilized to establish that the four groups followed normal distribution. Levene’s statistic test of homogeneity of variance was used to assess the equality of variances. VAS score data were then statistically analyzed with a repeated measures ANOVA test and unpaired *t*-test using Statistical Product and Service Solutions version 15 software (SPSS Inc., Chicago, IL, USA). Inter-group comparison for various stimuli was made using the post hoc Bonferroni test. A *p*-value of <0.05 was considered statistically significant.

## 3. Results

The repeated-measures ANOVA test showed a significant difference between different time periods, one day after the restoration, one week after the restoration, and then one month after the restoration for the cold, hot, and sweet stimuli (Table 2).

Inter-group comparison of mean scores for the different stimuli for different time periods was made using the post hoc Bonferroni test. A significant decrease in sensitivity after one day to after one week and after one month from restoration was observed for cold and sweet stimuli, with no significant decrease between one week and one month. However, for hot stimuli, a significant decrease was noted between the first day and month, and the first week and month, though no significant decrease was seen from the first day to the first week (Table 3).

An unpaired *t*-test was used, and no significant difference was observed between the maxillary and mandibular teeth at different periods of time and for various stimuli (Table 4).

A similar comparison was made for the different age groups. There was no significant difference between the age groups for the three stimuli, and there was no significant variation at various periods of time.

The inter-group comparison by post hoc Bonferroni test found that when the three desensitizers were analyzed for cold stimuli, Gluma showed a significant decrease compared to the Shield Force Plus and Telio CS desensitizers by the end of the first week and one month (Table 5).

Table 6 shows the inter-group comparison by post hoc Bonferroni test for hot stimuli, which showed a significant decrease when Gluma was compared to Shield Force Plus desensitizer by the end of the first week and one month. Moreover, Sheild Force Plus displayed a significant decrease after one week and one month when compared to the control group, where no desensitizer was used. Sheild Force Plus also showed a significant decrease after one week when compared to Telio CS.

Table 7 shows the inter-group comparison using a post hoc Bonferroni test of when the three desensitizers were analyzed for sweet stimuli. Gluma showed a significant decrease compared to the Shield Force Plus and Telio CS desensitizers by the end of the first week and one month.

## 4. Discussion

Composite is one of the most common esthetic restorative materials. Nevertheless, its biological risk is still being researched. Due to the synthetic components of the composite, pulpal integrity might be affected, leading to sensitivity after the restoration [5,6,10]. The constituents of composite have been associated with cytotoxicity in in vitro studies. They have been shown to influence cytokine mediation and also mitochondrial activity in the fibroblasts in vivo [11,12,13]. Tronstad stated that there is enough evidence to show that pulp and dentin act as a single tissue, and there are various mechanisms explaining this conclusion. The most accepted is the fluid movement in the dentinal tubules by hydrodynamics [14]. During the cavity preparation, thesehydrodynamics are affected, which might lead to postoperative sensitivity [15]. In the study by Christensen, he stated the significance of the liner to prevent this issue [16]. Many liners and other desensitizers are available today that claim to lower post-operative sensitivity [17,18]. However, many of the studies reporting the same findings were in vitro. Hence, in the present study, we compared the effect of the three desensitizers, Gluma, Shield Force Plus, and Telio CS, in an in vivo clinical trial at one day, after one week, and after one month. The VAS was employed to evaluate the subjective perception of post-operative sensitivity for hot, cold, and sweet stimuli. A 0–10 numerical score was considered, which is a common method of scoring pain.

It was observed that with time, sensitivity to cold drinks, hot drinks, and sugar decreased. This agrees with the study of Dondi et al. [19], where it was also observed that Gluma significantly reduced sensitivity in the test group. However, in the clinical trial performed by Sobral, no significant difference was observed with the application of Gluma for all three stimuli [20]. The present study was the first of its kind to compare the different age groups and the maxillary and mandibular teeth, although no significant variation was observed for the different time periods and the stimuli. It was noted that Gluma showed a better efficacy than the other two desensitizers for all the stimuli at different periods of time. This observation is in accordance with the study of Patil et al. [21], where they found a decrease in tooth sensitivity with the application of Gluma. The difference in the level of effectiveness among the three desensitizers could be due to the differences in their composition and mechanisms of action [8,9]. Gluma and Sheild Force Plus, are both HEMA-based desensitizers, and showed better results than Telio CS. The double coating effect of Shield Force desensitizer could be a possible reason of its better outcome in terms of decreasing dentin hypersensitivity, when compared to Telio CS desensitizer. The present study contradicts the study by Chandra et al. [22], which concluded that Shield Force Plus reduced the sensitivity more than Gluma by better occluding the dentinal tubules. In recent research, Moosavi et al. [23] suggested low-level laser therapy (LLLT) before placement of the resin composite to reduce post-operative sensitivity in Class V restorations. In the present study, all the groups treated with desensitizers showed significantly reduced pain levels, except the control group; thus, rejecting the null hypothesis.

This is one of the first studies to compare the three desensitizers in a randomized controlled clinical trial. Post-operative sensitivity after the composite restoration is yet to be thoroughly deciphered. One limitation of the present study is the duration. It was followed up for 1 month and was conducted with Class Ι cavity design. Previous studies have shown cytopathic effects with glutaraldehyde and HEMA, questioning the biocompatibility of the desensitizer agent components [24,25]. The cytotoxic potential of the desensitizers must be studied in further research. Age and gender effects must be re-evaluated with a larger sample size and stratified sampling. Furthermore, long-term, multi-centered clinical trials should be carried out in the future to reinforce the present results.

## 5. Conclusions

For the success of dental composite restoration, management of post-operative sensitivity is important. Gluma desensitizer showed better acceptance than the Shield Force and Telios CS in the present study. It is proposed that a longer follow-up with different designs should be implemented to understand the efficacy of desensitizers.

## Figures and Tables

**Figure 1 polymers-14-01417-f001:**
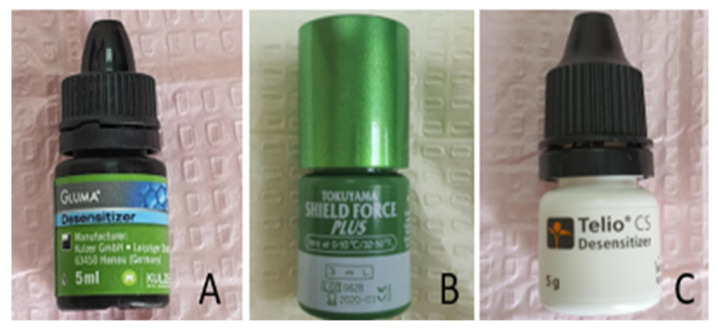
(**A**) Gluma Desensitizer (Heraeus Kulzer, Hanau, Germany), (**B**) Shield Force Plus Desensitizer (Tokuyama Dental America Inc., San Diego, CA, USA), and (**C**) Telio CS Desensitizer (Ivoclar Vivadent, Schaan, Liechtenstein).

**Figure 2 polymers-14-01417-f002:**
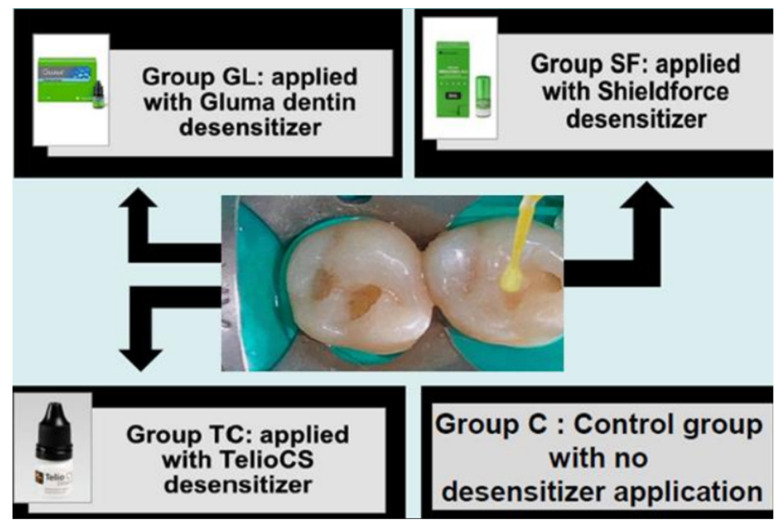
The 4 groups, Group GL: Gluma desensitizer applied before the restoration, Group SF: Shield Force Plus desensitizer applied before the restoration, Group TC: Telio CS desensitizer applied before the restoration, and Group C: no desensitizer applied before the restoration (control group).

**Figure 3 polymers-14-01417-f003:**
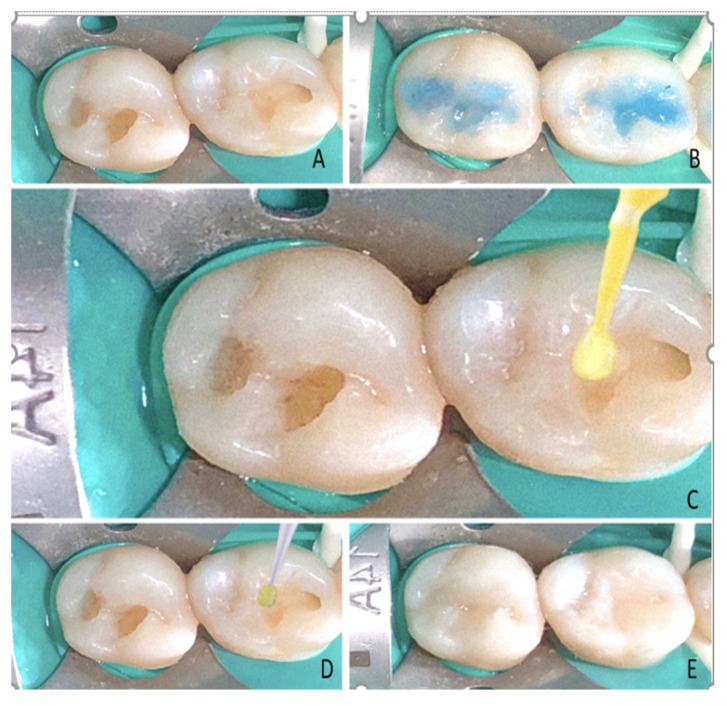
(**A**) Standard Class I cavity preparation, (**B**) acid-etched using 37% phosphoric acid, (**C**) desensitizer application performed with a disposable brush, according to the manufacturers’ instructions, (**D**) two layers of adhesive bonding agent applied and light-cured, (**E**) Tetric N-Ceram Bulk Fill (Ivoclar Vivadent) applied and condensed.

**Table 1 polymers-14-01417-t001:** The manufacturer details, composition, and mechanisms of action of the desensitizers and the composite used.

Group	Material Trade Name	Batch No.	Manufacturer	Composition	Mechanism of Action
2	Gluma dentin desensitizer	K010514	Heraeus Kulzer, Hanau, Germany	5% glutaraldehyde and 35% 2-hydroxyethyl methacrylate (HEMA) in an aqueous solution.	Glutaraldehyde reacts with the dentinal protein component to create a clogging mass which decreases the tubular size.
3	Telio CS desensitizer	Y09693	IvoclarVivadent, Schaan, Liechtenstein	35% polyethylene glycol dimethacrylate, 50% glutaraldehyde, 55% water, <0.01% maleic acid.	Both polyethylene glycol dimethacrylate (PEG-DMA) and glutaraldehyde, compose ideal material options to aid in sealing the dentinal tubules.
4	Shield Force Plus desensitizer	140E48	Tokuyama Dental America Inc., San Diego, CA, USA	10–30% 2-hydroxyethyl methacrylate (HEMA), 10–30% bisphenol A dis (2-hydroxy propoxy) dimethacrylate, 10–30% phosphoric acid monomer, 30–60% propan-2-ol, 5–10% triethylene glycol dimethacrylate, 5–10% water.	Direct reaction between the adhesive monomer with calcium component in the tooth to make the first coating layer. Another long-lasting layer is formed by curing.
	Ivoclar Tetric N Flow Flowable Bulk Fill Composite IVA	X19199	IvoclarVivadent, Schaan, Liechtenstein	Monomer matrix: 28% monomethacrylates and dimethacrylates Fillers: 71%-barium glass, ytterbium trifluoride, and copolymers Additives, initiators, stabilizers, and pigments (<1.0 wt%)	It can be light-cured in large increments of up to four millimeters, requiring only short light exposure times. The patented light activator Ivocerin is responsible for ensuring complete cure of the filling.

**Table 2 polymers-14-01417-t002:** Comparison of the VAS at different time periods for the various stimuli using the repeated-measures ANOVA test.

Stimuli	Mean	Standard Deviation	*F*-Value	*p*-Value
Cold				
After 1 day	2.41	2.52	62.047	<0.001 *
After 1 week	1.88	2.45		
After 1 month	1.75	2.37		
Hot				
After 1 day	0.31	1.26	5.820	0.018 *
After 1 week	0.30	1.24		
After 1 month	0.38	1.39		
Sweet				
After 1 day	1.06	2.11	16.665	<0.001 *
After 1 week	0.75	1.84		
After 1 month	0.73	1.83		

* statistically significant.

**Table 3 polymers-14-01417-t003:** Inter-group comparison of the VAS at different time periods for various stimuli using a post-hoc Bonferroni test.

Stimuli		Mean Difference	*p*-Value
Cold			
After 1 day	After 1 week	0.54	0.020 *
After 1 day	After 1 month	0.66	0.005 *
After 1 week	After 1 month	0.13	0.596
Hot			
After 1 day	After 1 week	0.01	1.000
After 1 day	After 1 month	−0.06	0.049 *
After 1 week	After 1 month	−0.08	0.046 *
Sweet			
After 1 day	After 1 week	0.31	0.041 *
After 1 day	After 1 month	0.34	0.035 *
After 1 week	After 1 month	0.03	0.961

* statistically significant.

**Table 4 polymers-14-01417-t004:** Comparison of the VAS for the location of the teeth for the various stimuli using an unpaired *t*-test.

	Maxillary		Mandibular		Mean Difference	*t*-Test Value	*p*-Value
Stimuli	Mean	Standard Deviation	Mean	Standard Deviation			
Cold							
After 1 day	1.95	2.56	2.57	2.51	−0.62	−0.95	0.347
After 1 week	1.15	2.18	2.12	2.50	−0.97	−1.54	0.128
After 1 month	1.15	2.18	1.95	2.41	−0.80	−1.31	0.193
Hot							
After 1 day	0.00	0.00	0.42	1.44	−0.42	−1.29	0.202
After 1 week	0.20	0.89	0.33	1.34	−0.13	−0.42	0.679
After 1 month	0.20	0.89	0.43	1.52	−0.23	−0.65	0.519
Sweet							
After 1 day	0.55	1.70	1.23	2.21	−0.68	−1.26	0.211
After 1 week	0.30	1.34	0.90	1.96	−0.60	−1.27	0.208
After 1 month	0.30	1.34	0.87	1.96	−0.57	−1.20	0.234

**Table 5 polymers-14-01417-t005:** Inter-group comparison of the VAS at different time periods for cold stimuli by post hoc Bonferroni test.

Cold Stimuli	First Group	Second Group	Mean Difference	*p*-Value
After 1 day	Group C	Group G	−1.15	0.914
	Group C	Group SF	−1.40	0.494
	Group C	Group TC	−0.70	1.000
	Group GL	Group SF	−0.25	1.000
	Group GL	Group TC	0.45	1.000
	Group SF	Group TC	0.70	1.000
After 1 week	Group C	Group G	1.30	0.432
	Group C	Group SF	−1.50	0.231
	Group C	Group TC	−0.90	1.000
	Group GL	Group SF	−2.80	0.001 *
	Group GL	Group TC	−2.20	0.017 *
	Group SF	Group TC	0.60	1.000
After 1 month	Group C	Group G	1.30	0.408
	Group C	Group SF	−1.00	0.951
	Group C	Group TC	−0.90	1.000
	Group GL	Group SF	−2.30	0.010 *
	Group GL	Group TC	−2.20	0.015 *
	Group SF	Group TC	0.10	1.000

* statistically significant.

**Table 6 polymers-14-01417-t006:** Inter-group comparison of the VAS at different time periods for hot stimuli using a post-hoc Bonferroni test.

Hot Stimuli			Mean Difference	*p*-Value
After 1 day	Group C	Group G	−0.35	1.000
	Group C	Group SF	−0.90	0.138
	Group C	Group TC	0.00	1.000
	Group GL	Group SF	−0.55	0.962
	Group GL	Group TC	0.35	1.000
	Group SF	Group TC	0.90	0.138
After 1 week	Group C	Group G	−0.10	1.000
	Group C	Group SF	−0.90	0.026 *
	Group C	Group TC	−0.20	1.000
	Group GL	Group SF	−0.80	0.037 *
	Group GL	Group TC	−0.10	1.000
	Group SF	Group TC	0.70	0.042 *
After 1 month	Group C	Group G	−0.10	1.000
	Group C	Group SF	−1.20	0.033 *
	Group C	Group TC	−0.20	1.000
	Group GL	Group SF	−1.10	0.044 *
	Group GL	Group TC	−0.10	1.000
	Group SF	Group TC	1.00	0.119

* statistically significant.

**Table 7 polymers-14-01417-t007:** Inter-group comparison of the VAS at different time periods for sweet stimuli using a post hoc Bonferroni test.

Sweet Stimuli			Mean Difference	*p*-Value
After 1 day	Group C	Group G	−0.35	1.000
	Group C	Group SF	−0.60	1.000
	Group C	Group TC	−0.90	1.000
	Group GL	Group SF	−0.25	1.000
	Group GL	Group TC	−0.55	1.000
	Group SF	Group TC	−0.30	1.000
After 1 week	Group C	Group G	0.40	1.000
	Group C	Group SF	−0.70	1.000
	Group C	Group TC	−0.70	1.000
	Group GL	Group SF	−1.10	0.035 *
	Group GL	Group TC	−1.10	0.035 *
	Group SF	Group TC	0.00	1.000
After 1 month	Group C	Group G	0.50	1.000
	Group C	Group SF	−0.70	1.000
	Group C	Group TC	−0.70	1.000
	Group GL	Group SF	−1.20	0.038 *
	Group GL	Group TC	−1.20	0.038 *
	Group SF	Group TC	0.00	1.000

* statistically significant.

## Data Availability

The data that support the findings of this study are available from the corresponding author upon reasonable request.

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
