# Peer review of "Efficacy of Three Commercially Available Desensitizers in Reducing Post-Operative Sensitivity Following Composite Restorations: A Randomized Controlled Clinical Trial"

_polymers, 2022, doi:10.3390/polym14071417_

Round 1

Reviewer 1 Report

Manuscript ID: polymers-1659204

Dear. Authors,

This topic was evaluated with VAS analysis was very interesting for dentists.

Because young patients are mainly evaluated in this clinical trial.

There are several issues that should be addressed in the manuscript before further consideration for publication.

  1. Materials and Method,

A clinically significant reduction was considered using difference of 1 in the VAS.

Why did you determine difference of 1?   Please mention if there are any references.

  1. Materials and methods

2.1 This following formula was used to calculate the sample size.

N = [4 × S2 (Zα + Zβ)2]/(d)2    Please mention if there are any references.

Table1.

Group 2, 3 and 4 are mentioned in Table1. Where is group 1?

2.5 Evaluation of post-operative sensitivity

The repeated measures ANOVA and t-test are used for data with normal distribution and equal of variances.

Normality test and homogeneity test should be mentioned in this section.

Data is statistically analyzed by repeated measures ANOVA and t-test.

You should add “software Ver., companies, head offices, city or state and Country” in manuscript.

  1. Discussion

Age and gender effect must be re-evaluated in future study.

However, post-operative sensitivity (more than 45 years) might be decreased than young patients because dental pulp of old patients are smaller than that of young patients. As a result, there is no significant difference among desensitizers.

You should discuss about above reason comparing with your references if it is possible.

Reviewer 2 Report

Authors in this paper evaluated and compared the effectiveness of three desensitizers agents (Gluma, Shield Force Plus and Telio CS) in reducing post-treatment sensitivity for Class I composite restorations. Clinical trial results showed that Gluma desensitizer showed better acceptance than the Shield Force and Telios. The data presented in the manuscript basically support the author's point of view. After appropriate revisions, I recommend the manuscript to be accepted. The main points are as follows:

  1. Why choose these three desensitizers for the research? In addition to the two cited references, it is recommended to introduce them in the introduction section.
  2. In the introduction section, the author used the phrase "the same three desensitizing agents", please check if the wording is incorrect, if not, please explain why "the same".
  3. In Table 2, compared to the significant differences of the cold and sweet stimuli, differences in hot stimulation do not appear to be significant. In addition, in the full paper, there is no big difference in hot stimulation, whether it is meaningful to discuss hot stimulation in this case.
  4. The contents of Tables 6 and 7 are grouped in a mess, and it is recommended to modify the format as in Table 5.
  5. In the paper, many of the data in the table are not statistically significant. Can you explain this situation in detail?
  6. There are many tables in the text, while few descriptions and discussions of the tables. It is recommended to add discussion of the tables as appropriate
